# Prediction of Maximum Pressure at the Roofs of Rectangular Water Tanks Subjected to Harmonic Base Excitation Using the Multi-Gene Genetic Programming Method

**Iman Bahreini Toussi [1,*]**, **Abdolmajid Mohammadian [1]** and **Reza Kianoush [2]**

1   Department of Civil Engineering, University of Ottawa, 161 Louis Pasteur Private,
    Ottawa, ON K1N 6N5, Canada; amohamma@uottawa.ca
2   Department of Civil Engineering, Ryerson University, 350 Victoria Street, Toronto, ON M5B 2K3, Canada;
    kianoush@ryerson.ca
*   Correspondence: ibahr094@uottawa.ca; Tel.: +1-(613)-562-5800 (ext. 6159)

**Abstract:** Liquid storage tanks subjected to base excitation can cause large impact forces on the tank roof, which can lead to structural damage as well as economic and environmental losses. The use of artificial intelligence in solving engineering problems is becoming popular in various research fields, and the Genetic Programming (GP) method is receiving more attention in recent years as a regression tool and also as an approach for finding empirical expressions between the data. In this study, an OpenFOAM numerical model that was validated by the authors in a previous study is used to simulate various tank sizes with different liquid heights. The tanks are excited in three different orientations with harmonic sinusoidal loadings. The excitation frequencies are chosen as equal to the tanks' natural frequencies so that they would be subject to a resonance condition. The maximum pressure in each case is recorded and made dimensionless; then, using Multi-Gene Genetic Programming (MGGP) methods, a relationship between the dimensionless maximum pressure and dimensionless liquid height is acquired. Finally, some error measurements are calculated, and the sensitivity and uncertainty of the proposed equation are analyzed.

**Keywords:** liquid storage tanks; base excitation; artificial intelligence; Multi-Gene Genetic Programming; computational fluid dynamics; finite volume method

## 1. Introduction

Earthquakes cause damage to various types of structures, and buildings, dams, reservoirs, and liquid storage tanks may be victims of an earthquake excitation. Sloshing in a liquid storage tank can cause irreversible structural failure and spillage of the liquid material into the environment, and this liquid, if toxic or flammable, may affect the area for a long time, even permanently. Thus, protecting liquid storage tanks from damage during an earthquake is crucial. One of the causes is related to the pressure exerted on the roof of the tank due to the sloshing of the liquid. Therefore, it is necessary for a designer to know the maximum pressure caused by such effects on a tank's roof.

Analytical, numerical, and experimental solutions have been introduced by various scholars. Housner [1] provided an analytical solution that is adopted in some design codes and standards such as the ACI 350.3 from the American Concrete Institute [2]. Housner's method divides the liquid into two parts, i.e., impulsive and convective. The former is the lower part of the liquid that moves in unison with the tank walls, while the latter is the upper part of liquid that creates sloshing in a tank. The impulsive mass is assumed to be rigidly connected to the tank's walls, while the convective mass is modeled by a mass–spring system. Figure 1 illustrates Housner's model for ground-supported tanks. Despite attempts at developing analytical solutions other than Housner's method (e.g., Isaacson [3]), most previous studies have concentrated on numerical analyses. The

goal of such studies is to provide a solution to the Navier–Stokes equations given in Equations (5)–(8), which are the governing equations in fluid flow. Cho and Cho [4] developed a combined finite element–boundary element (FE–BE) method to predict liquid behavior and its interaction with a structure, and Liu and Lin [5] studied a numerical model to solve 3D non-linear sloshing in a liquid storage tank. Their model adopted the volume of fluid (VOF) method for tracking a free surface in conjunction with the finite difference method (FDM). Chen et al. [6] formulated a numerical model that is based on Reynolds-averaged Navier–Stokes (RANS) fluid motion, which proved to be in good agreement with the experimental data from Daewoo Shipbuilding & Marine Engineering Co., Ltd. (DSME) [7]. The data were obtained from tests on a rectangular tank with plan dimensions of 800 mm × 400 mm and a height of 500 mm that was horizontally excited with different frequencies.

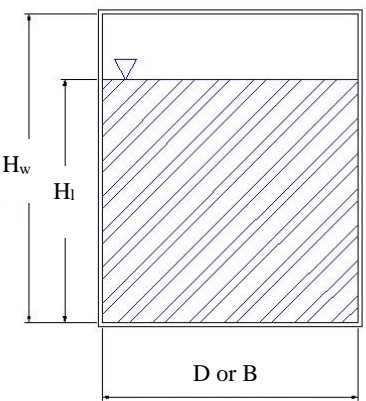 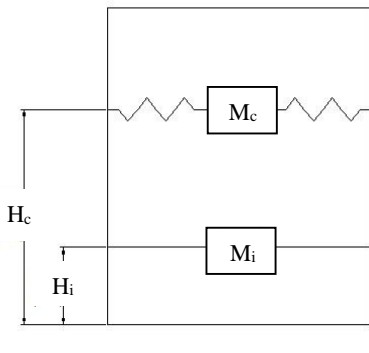

**Figure 1.** Schematic view of Housner's simplified model.

In recent years, artificial intelligence (AI) has been evolving in all aspects of human life, including engineering problems (Afan et al. [8]). There are several methods for the estimation of a relationship between scattered data based on AI. Among them, the group method of data handling (GMDH; Ivakhnenko and Ivakhnenko [9]) and evolutionary polynomial regression (EPR) can be mentioned. AI techniques such as support vector machine (SVM), artificial neural networks (ANNs), adaptive neuro-fuzzy inference system (ANFIS), Genetic Programming (GP) have recently been used for engineering problems such as water quality index and groundwater level modeling (e.g., Mohammadpour et al. [10]; Ghani and Azamathulla [11]; He et al. [12]; Lallahem et al. [13]; Daliakopoulos et al. [14]; Mirzavand et al. [15]; and Mohammadpour et al. [16]).

Model tree (MT)—a sub-class of the regression tree method—is another regression method in which an equation is generated at each node [17]. In a regression tree, a constant or a relatively simple regression model is used to demonstrate the data [18]. A genetic based method known as GP is also used for the regression of data. In this method, a set of sub-trees is randomly generated based on user-defined specifications using arithmetic operators (i.e., +, −, ×, /), non-linear functions (e.g., sin, cos, log), etc. [19]. The goal is to minimize the errors (e.g., root mean square error (RMSE)) in newer generations until an acceptable error is reached.

Another method for formulating scattered data based on AI is gene expression programming (GEP), which was introduced by Ferreira in 1999 (Sattar and Gharabaghi [20]). This method can be employed to develop relationships between data with minimal error [21]. Azamathulla [22] adopted this method to estimate the scour depth downstream of sills. To do so, he used the following procedure: (1) choose a fitness function; (2) choose a set of terminals (T) and functions (F) to shape chromosomes; (3) choose the chromosome architecture (i.e., head length and the number of genes); (4) choose the linking function (e.g., addition and multiplication operators); and (5) choose the set of genetic operators (e.g., mutation, transportation, etc.). He compared his results with the equa-

tion obtained by Chinnarasri and Kositgittiwong [23], which at the time had the lowest error value, and found that the proposed equation using the GEP model had a higher accuracy. Najafzadeh et al. [24] used three methods, i.e., GEP, MT, and EPR, to predict the maximum scour depth near piers with debris accumulation. Gholami et al. [25] used the GEP method to predict the characteristics of stable bank channels. They obtained their own experimental data as well as data from previous experimental studies to complete their GEP modelling. The results were compared with available theoretical and experimental methods. Despite a good agreement and accuracy, the model's complexity was found to be higher in comparison with older analytical methods, and therefore the GEP method was not suggested by the authors. Sheikh et al. [26] applied GEP to analyze shear stress distribution in circular channels with flat beds subject to sediment deposition. They proposed equations for predicting the base shear applied to the bed and the walls of such channels. It was found that the GEP model could lower errors and uncertainties, and hence the model was recommended for the base shear analysis of circular channels with flat beds.

A sub-class of the GP method known as Multi-Gene Genetic Programming (MGGP) can be used in problems with higher complexity. A gene is a weighted linear combination of outputs from a GP tree. In this method, the user has control over the maximum number of genes and the depth of the model tree [27]. In this method, multiple genes are combined to produce an MGGP model. AI techniques have shown to be capable of accurate prediction, and with the development of computing systems, they have become easier to use. However, to the best of the authors' knowledge, they have not been employed in the prediction of pressures and forces in water tanks. Previous studies in engineering applications have shown promising results for MGGP in comparison with other AI techniques such as ANN, ANFIS, traditional GP, etc. (Kaydani et al. [28]; Safari and Mehr [29]; Mehr and Nourani [30]).

The use of GP methods in civil engineering is becoming increasingly popular. Gandomi et al. [31] proposed an empirical model for predicting the ultimate shear strength of reinforced concrete (RC) deep beams using GEP. The results were compared with design codes such as ACI and CSA, and the model was found to give better results than the design codes when compared to the available experimental and numerical data. Gandomi et al. [32] developed a model to find the shear capacity of RC beams without stirrups using the GEP method. To avoid overfitting, they divided the data into three groups of learning, validation, and testing on a random basis. The developed model was tested against the available data and several design codes (e.g., ACI, CSA, NZS, etc.) for various sizes and models of RC beams and was found to give compatible results. GEP can be used in various fields of civil engineering as an optimization method. Zahiri et al. [33] investigated the applications of GEP in hydraulic engineering and found it applicable in different areas, such as estimation of scour depth, discharge rate, and land transport in rivers.

In the present study, data generated by a validated OpenFOAM (Open-Source Field Operation and Manipulation) [34] model are used. The maximum pressure on the roof of a tank is the parameter of interest. Several tank sizes with various liquid heights are excited by a resonance frequency, and the maximum hydrodynamic pressure at the roof of the tank in each case is obtained. Using the GP method in both Single-Gene and Multi-Gene modes, an equation is proposed for predicting the maximum pressure at the roof of the tank. Finally, the proposed equation's reliability is investigated and discussed through error measurements as well as uncertainty and sensibility analyses.

To the best of the authors' knowledge, a study such as this one that predicts the maximum pressure at the roof of a liquid storage tank subjected to base excitation has not been addressed previously. The design codes generally provide a minimum free-board, and if the provided free-board is not sufficient, it is left to the designers to decide how to design the roof. No further data are provided in that manner in the design codes. Furthermore, previous studies have not investigated the pressures at the roof of the tank with the intention of finding a relationship between the tank size and the maximum

pressure on the roof. The available codes and standards do not provide details for designing the roof of tanks with insufficient freeboard, and they only recommend designing the roof to resist uplift pressures. Therefore, this study can provide a good estimate of those pressures and help with the design process.

The results from this study can help provide empirical formulations to appropriately estimate the hydrodynamic pressures at the roof of a liquid tank subjected to base excitations. This can be adopted in design codes and standards to better address the uplift forces and hydrodynamic pressures at the roof level. In addition, the artificial intelligence component of this research can significantly reduce computational cost and time.

Although earthquake and harmonic excitations have different characteristics, it was found in a previous study [35] that harmonic resonance excitations can produce higher hydrodynamic pressures on the roof of a tank compared to earthquake excitations, which is the reason this kind of loading was applied in this study instead of earthquake excitations

This paper is organized as follows. Section 2 deals with the details and equations of numerical modeling and MGGP. Section 3 presents the results, discussions, and error measurements, and some concluding remarks complete the study.

## 2. Materials and Methods

### 2.1. Numerical Modelling

An OpenFOAM model was previously developed and validated by the authors [35]. The same model was used to generate data for the current study. The maximum hydrodynamic pressure at the roof of rectangular tanks is the parameter of interest in this study. Hence, pressure sensors were distributed on one quarter of the roof for each simulation.

Four different tank sizes were used in the study, the dimensions of which are presented in Table 1. For each tank, a minimum of six different liquid heights were simulated, as discussed later. Since the direction of an earthquake cannot be predicted, four different tank orientations were tested, and among them, the highest roof pressure for each liquid height in each tank was found.

**Table 1.** Dimensions of tanks used in the study.

|  | Length (mm) | Width (mm) | Height (mm) |
|---|---|---|---|
| Size 1 | 755 | 300 | 300 |
| Size 2 | 1978 | 779 | 1200 |
| Size 3 | 1283 | 327 | 1200 |
| Size 4 | 683 | 342 | 1200 |

Many previous studies (e.g., [4,36,37]) have shown that Housner's simplified method [1] predicts resonance frequency accurately, and hence in this study the same method was applied.

Based on Housner's method, the resonance frequency in a rectangular tank can be calculated as follows:

$$M_c = M \frac{\tanh 1.7 \, L/h}{1.7 \, L/h} \tag{1}$$

$$k_c = 3 \frac{M_1^2}{M} \frac{gh}{L^2} \tag{2}$$

$$\omega_c = \sqrt{\frac{k_c}{M_c}} \tag{3}$$

$$T_c = \frac{2\pi}{\omega_c} \tag{4}$$

where $M_c$ is the mass of the convective part of the liquid ($c$ = convective), $M$ is the total liquid mass, $L$ is half of the tank length, $h$ is the total liquid height, $k_c$ is the stiffness of the assumed spring that connects the convective mass to the tank's walls in the direction of

movement, $g$ is ground acceleration equal to 9.81 m/s$^2$, and $\omega_c$ and $T_c$ are the resonance frequency and resonance period of the first (fundamental) mode of the oscillating liquid, respectively. In lieu of Housner's method to determine the natural frequency of the tank, Lamb's formula can be used for simplicity [38]. In Table 2, the resonance frequencies that were applied to each tank based on the size and liquid height are presented. Each tank size–liquid height combination was simulated at four different orientations of 0°, 30°, 60°, and 90°. Since the direction of an earthquake is not predictable, the maximum pressure among all orientations was used as the input for the GP section. In other words, the maximum of maximums was found and applied to the GP. The excitation orientations of 0°, 30°, and 60° are presented in Figure 2.

**Table 2.** Frequency applied to each tank based on the tank size and liquid height.

| | Length | Width | Tank Height | Liquid Height | Dimensionless Liquid Height | $\omega_i$ | $T_i$ |
|---|---|---|---|---|---|---|---|
| | (mm) | (mm) | (mm) | (mm) | ($h_l$/L) | (rad/s) | (s) |
| | | | | 100 | 0.265 | 4.023 | 1.562 |
| | | | | 120 | 0.318 | 4.354 | 1.443 |
| | | | | 145 | 0.384 | 4.705 | 1.335 |
| | | | | 200 | 0.53 | 5.282 | 1.190 |
| | | | | 230 | 0.609 | 5.510 | 1.140 |
| | | | | 250 | 0.662 | 5.636 | 1.115 |
| | | | | 280 | 0.742 | 5.792 | 1.085 |
| | | | | 1100 | 1.112 | 3.819 | 1.645 |
| | | | | 1000 | 1.011 | 3.777 | 1.663 |
| Size 2 | 1978 | 779 | 1200 | 900 | 0.910 | 3.721 | 1.689 |
| | | | | 800 | 0.809 | 3.644 | 1.724 |
| | | | | 700 | 0.708 | 3.540 | 1.775 |
| | | | | 600 | 0.607 | 3.400 | 1.848 |
| | | | | 1100 | 1.714 | 4.858 | 1.293 |
| | | | | 1000 | 1.559 | 4.845 | 1.297 |
| | | | | 900 | 1.403 | 4.824 | 1.303 |
| | | | | 800 | 1.247 | 4.789 | 1.312 |
| | | | | 700 | 1.091 | 4.732 | 1.328 |
| | | | | 600 | 0.935 | 4.640 | 1.354 |
| | | | | 1100 | 3.221 | 6.686 | 0.940 |
| | | | | 1000 | 2.928 | 6.686 | 0.940 |
| Size 4 | 683 | 327 | 1200 | 900 | 2.635 | 6.685 | 0.940 |
| | | | | 800 | 2.343 | 6.682 | 0.940 |
| | | | | 700 | 2.05 | 6.677 | 0.941 |
| | | | | 600 | 1.757 | 6.662 | 0.943 |

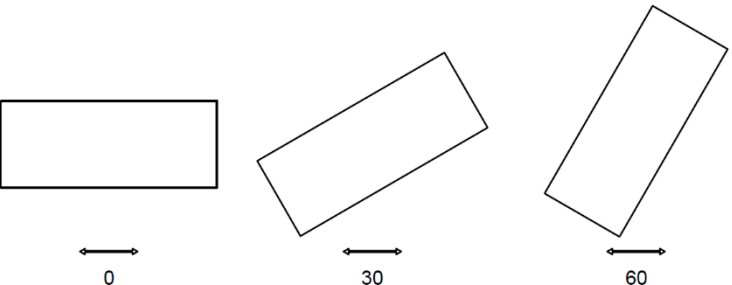

**Figure 2.** Tank orientations for simulations.

After finding the resonance frequency for each tank size and liquid height, numerical modelling was performed using OpenFOAM software. The OpenFOAM model can provide numerical solutions for various types of engineering problems, such as heat transfer, mass transport, liquid flow, etc. It can also solve fluid–structure interaction problems based on computational fluid dynamics (CFD) modelling [39]. Navier–Stokes equations in Equations (5)–(8) are solved for these types of problems.

$$\frac{\partial u}{\partial x} + \frac{\partial v}{\partial y} + \frac{\partial w}{\partial z} = 0 \tag{5}$$

$$\frac{\partial u}{\partial t} + u\frac{\partial u}{\partial x} + v\frac{\partial u}{\partial y} + w\frac{\partial u}{\partial z} = -\frac{1}{\rho}\frac{\partial p}{\partial x} + \nu\nabla^2 u \tag{6}$$

$$\frac{\partial v}{\partial t} + u\frac{\partial v}{\partial x} + v\frac{\partial v}{\partial y} + w\frac{\partial v}{\partial z} = -\frac{1}{\rho}\frac{\partial p}{\partial y} + \nu\nabla^2 v \tag{7}$$

$$\frac{\partial w}{\partial t} + u\frac{\partial w}{\partial x} + v\frac{\partial w}{\partial y} + w\frac{\partial w}{\partial z} = -\frac{1}{\rho}\frac{\partial p}{\partial z} + \nu\nabla^2 w - g \tag{8}$$

in which

$$\nabla^2 = \frac{\partial^2}{\partial x^2} + \frac{\partial^2}{\partial y^2} + \frac{\partial^2}{\partial z^2} \tag{9}$$

and $\rho$ and $p$ are the liquid density (kg/m$^3$) and total pressure (Pa) respectively; $u$, $v$, and $w$ are the particle speeds in the $x$, $y$, and $z$ directions (m/s); $t$ is time (s); and $g = 9.81$ m/s$^2$ is the gravity acceleration and

$$\rho = \alpha\rho_1 + (1 - \alpha)\rho_2 \tag{10}$$

where $\rho_1$ and $\rho_2$ are the densities of air and water, respectively, and $\alpha$ indicates the volume of each particle that is filled with each of the fluids. The value of $\alpha$ varies between 0.0 and 1.0, with 1.0 meaning the cell is filled with water and 0.0 indicating air. A value of 0.5 is allocated to the free surface. Any value between 0.0 and 0.5 indicates air, and a value between 0.5 and 1.0 indicates water.

Given the very high momentum of the flow, turbulent stresses have a negligible effect on the flow in comparison with the liquid sloshing forces, and hence, turbulence was not modeled in this study.

### 2.1.1. Computational Setup

- Mesh

In this study, a structured cubic mesh was used. By running a mesh sensitivity analysis, the optimum mesh size was found. To do so, the pressure at the top corner of the tank was measured with various mesh sizes.

- Initial conditions

For the initial conditions, the velocity, acceleration, and displacement fields were set to zero.

- Wall boundary conditions

The "no flow, frictionless" wall boundary condition is applied to the base and the side walls of the tank. This implicit boundary condition is used when no flow crosses the wall, and the shear stress at the wall and normal gradient of tangent velocity were set to zero. In other words, no fluid enters or exits the boundary where this condition is applied. This boundary condition is applied as follows:

$$U_n = 0 \tag{11}$$

$$\frac{\partial}{\partial n} U_\tau = 0 \tag{12}$$

where $U_n$ and $U_\tau$ are the normal and the tangential velocities of the flow, respectively, and $n$ is the normal vector of the boundary.

- Free surface boundary conditions

The pressure at the free surface is set to zero, and the free surface is modelled using the volume of fluid (VoF) method according to the following equation:

$$\frac{\partial \alpha}{\partial t} + \frac{\partial(\alpha u)}{x} + \frac{\partial(\alpha v)}{y} + \frac{\partial(\alpha w)}{z} = 0 \tag{13}$$

### 2.1.2. CFD Details

In the mesh sensitivity analysis, a mesh size of 6 mm × 6 mm × 6 mm was found to be reasonably accurate. An adjustable scheme was chosen for the time-step, with a maximum step size of 0.05 s. This means each time-step is chosen based on the previous step. This helps with the accuracy of the simulation results; however, it has higher computational costs.

In the validated OpenFOAM model, an eddy viscosity of $2 \times 10^{-4}$ m$^2$/s was found to provide the best results compared to the experimental data.

A total of eighteen pressure sensors (probes) are distributed on one quarter of the roof for each of the simulated tanks. The long duration of the simulations is expected that the pressure distribution on a quarter of the domain can be representative of the entire roof. In addition, in this study, the quarter of the roof with the highest pressure was selected for the GP analysis. The placement of sensors on the roof of the tank are presented in Figure 3. Using these sensors, the pressure distribution on the roofs of the tanks can be found. Figure 4 shows a sample of the CFD output; more details on the OpenFOAM model are given by Bahreini et al. [35].

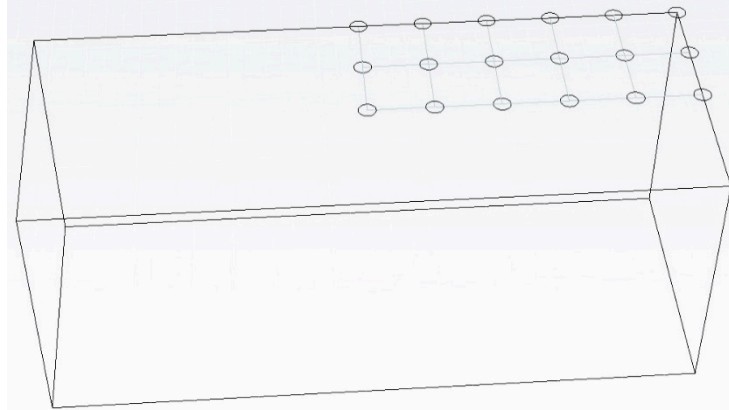

**Figure 3.** Sensor arrangement at the roof of the tank.

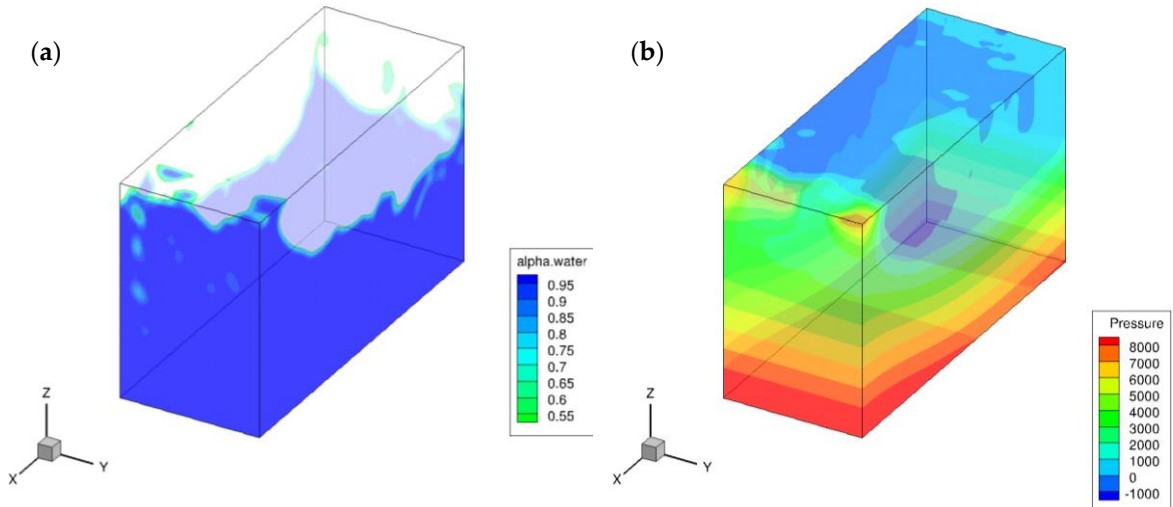

**Figure 4.** Computational fluid dynamics (CFD) outputs for tank size 2, with 800 mm water depth at $0°$ orientation and time $t = 9.50$ s; (**a**) liquid surface and (**b**) pressure.

### 2.2. Genetic Programming

Genetic programming (GP) is a method based on artificial intelligence that can be used in optimization problems. This method can be applied in Single-Gene and Multi-Gene models. In this method, the structure of the solution is not specified at the beginning and is shaped throughout the evolution [40]. Initial chromosomes are created, and during generations of evolutions and mutations, newer chromosomes with optimized characteristics are created. These cycles continue until the maximum number of iterations is reached or until the optimization reaches a point that is close to the solution (i.e., the error is negligible). In the Single-Gene model, mutations occur to one gene, while in the Multi-Gene method, there are mutations and crossovers across several genes.

In this method, the goal is to find the best-fit expression using the fit function (Equation (14)). This function has a value between 0 to 1000, with 1000 being the fittest, i.e., with the minimum error.

$$f_i = 1000 \frac{1}{1 + RRSE_i} \tag{14}$$

where

$$RRSE_i = \sqrt{\frac{\sum_{j=1}^{n} \left( P_{(ij)} - T_j \right)^2}{\sum_{j=1}^{n} \left( T_j - \overline{T} \right)^2}} \tag{15}$$

and $i$ is the number of the fit function, $j$ is the number of data, $P_{(ij)}$ is the calculated value for $j$th data based on $i$th function, $T_j$ is the actual value for the $j$th data, and $\overline{T}$ is the average of the $T_j$ values.

In GP-based methods, an initial gene or tree is randomly created, and the process starts. Several reproductions, including mutation (i.e., random changes in a gene and replacing a material with another material) and crossover (i.e., interchange of materials between the parent genes) operations, take place until the termination conditions are fulfilled.

Each gene is in a shape of a tree and consists of two types of nodes: (1) operator nodes, being mathematical operators (e.g., $+$, $-$, $\times$, $/$, power, sin, cos, log, etc.); and (2) operand nodes, which are the input variables, e.g., $x_1$, $x_2$, etc. (Pandey et al. [41]).

Here, an example is presented for further explanation and a better understanding. In a regression problem with two operands of $x_1$ and $x_2$ (i.e., $y = f(x_1, x_2)$, $y$ is dependent on two variables of $x_1$ and $x_2$), $A_1$ and $B_1$ are randomly created parent genes as follows:

$$A_1 = (2.3 \times x_1) - (sinx_2) \tag{16}$$

$$B_1 = \left(1.1 \times x_1^2\right) + (logx_2) \tag{17}$$

In a crossover process, a sub-tree of the parent gene $A_1$ is switched with a sub-tree of the parent gene $B_1$, resulting in second generation genes, $A_2$ and $B_2$:

$$A_2 = (2.3 \times x_1) - \left(1.1 \times x_1^2\right) \tag{18}$$

$$B_2 = (logx_2) + (sinx_2) \tag{19}$$

And in a mutation process, a sub-tree of each of the genes $A_2$ and $B_2$ is replaced by a new randomly chosen sub-tree, creating the third-generation genes, $A_3$ and $B_3$:

$$A_3 = (1.3 \times x_2^3) - \left(1.1 \times x_1^2\right) \tag{20}$$

$$B_3 = (logx_2) + \left(\frac{x_1}{x_2}\right) \tag{21}$$

This sequence continues until the termination conditions are fulfilled. At the end, the two genes are combined to form the equation:

$$Y_i = \alpha(A_i) + \beta(B_i) + C \tag{22}$$

which, in this three-generation example, is as follows:

$$y = \alpha\left[(1.3 \times x_2^3) - \left(1.1 \times x_1^2\right)\right] + \beta[(logx_2) + \left(\frac{x_1}{x_2}\right)] + C \tag{23}$$

where $\alpha$ and $\beta$ are called gene weights, and $C$ is a constant bias term. The gene weights and bias term are calculated by an ordinary least-squared method. Figure 5 shows the procedure of this example in the form of MGGP trees.

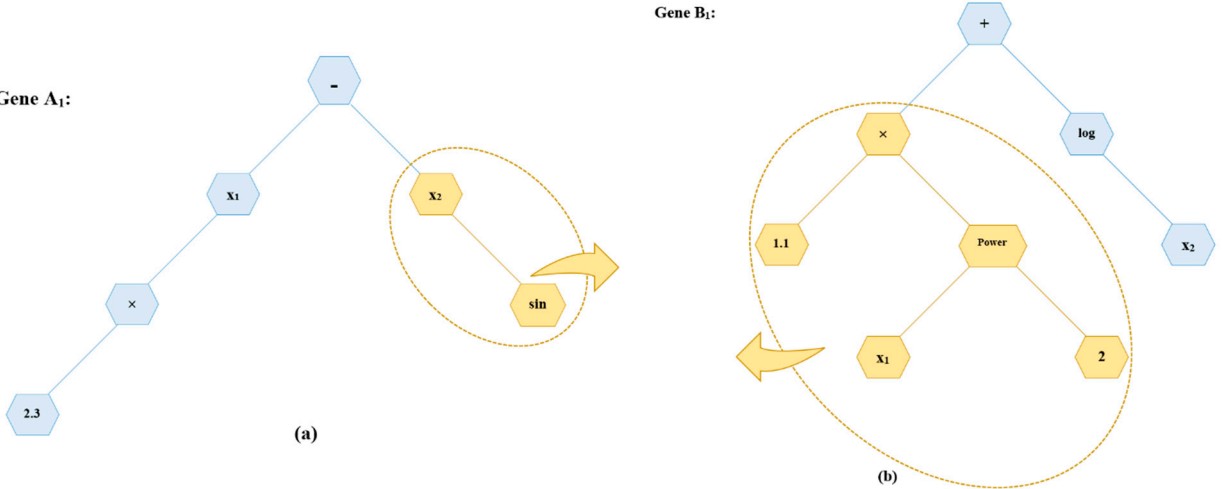

**Figure 5.** *Cont.*

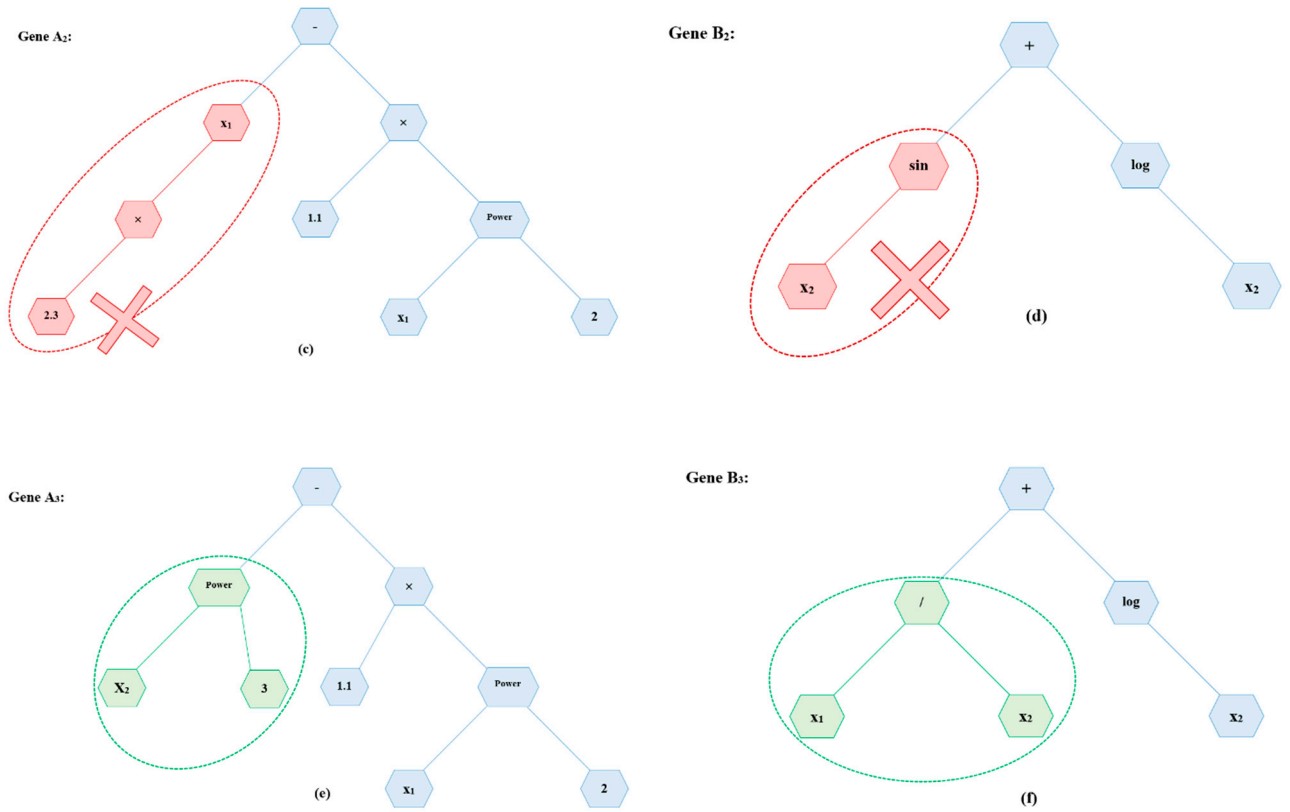

**Figure 5.** An example of a Multi-Gene Genetic Programming (MGGP) procedure.

In the current study, using MATLAB, an open-source MGGP algorithm (Genetic Programming Toolbox for the Identification of Physical Systems; GPTIPS) [42] is run to provide the general shape of the prediction function. In this algorithm, there is a random initial assumption for the function; then, the function is developed through generations until the error is minimized. Finally, using non-linear least squared optimization, an optimized equation is obtained that can be used for further analysis, as described in the following. This algorithm uses Pareto theory to find a balance between the fitness and complexity of the model in order to select the optimum model.

Figure 6 shows an example output tree of the MGGP algorithm. In this tree, the operators plus (+), minus (−), division (/), and multiplication (×) are used. The tree depth in this example is 12, and it has a total of 35 nodes.

In this method, chromosomes are introduced as computer programs of different shapes and sizes, with each consisting of sub-programs called genes, i.e., each chromosome is composed of genes. A typical GP method procedure is as follows:

1.   A set of variables is initiated.
2.   The chromosomes' architecture is defined.
3.   The chromosomes are randomly formulated.

This cycle continues until the function that best fits the data is found. For this study, from a total of 25 samples, 80% (20 samples) were used to train the model while 20% (5 samples) were used for testing (i.e., for validating the model). The trained data are expected to show higher accuracy and smaller errors since the model is directly obtained from this set of data. The tested and trained data are chosen on a random basis.

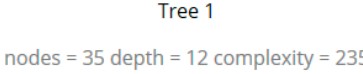

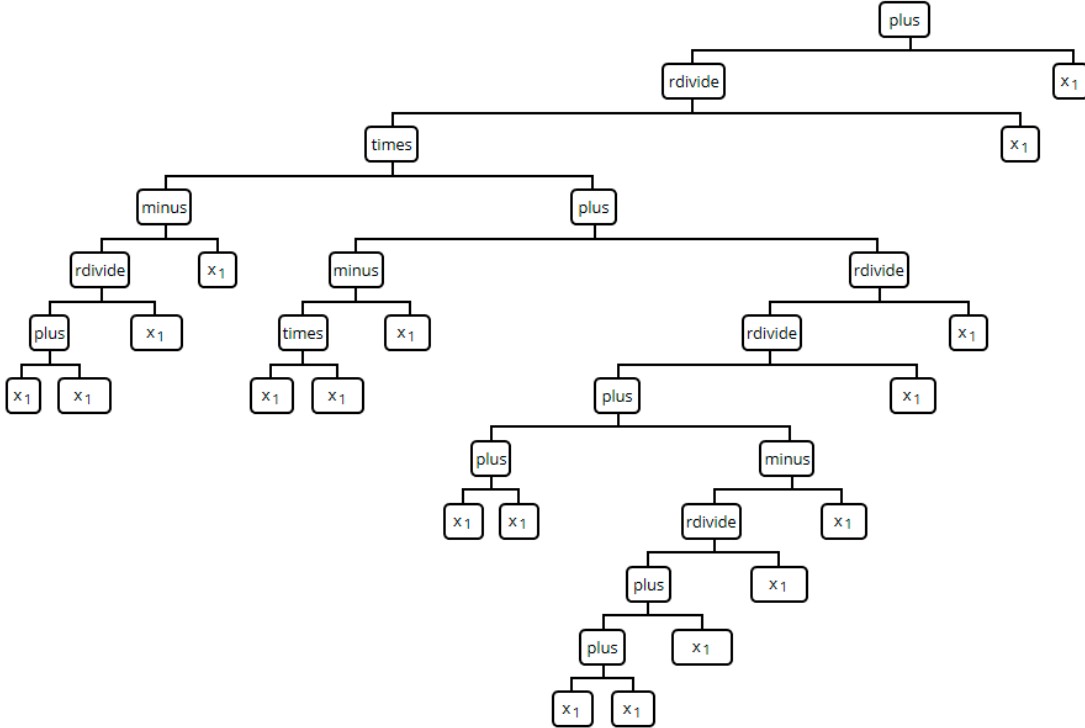

**Figure 6.** MGGP example output.

## 3. Results and Discussion

### 3.1. Numerical Modelling

Following the completion of the simulations, the results were analyzed for each case. At this stage, contours illustrating the maximum pressure distribution (not at a specific time-step but over the simulation time) on the roof are plotted for each simulation. Contours associated with the 755 mm × 300 mm tank are presented in Figures 7–9. In the figures, the bottom left represents the center of the roof with dimension of (0, 0), while the top right shows the corner (375.5, 150). The results from the numerical models show that in 46 out of 67 simulations (67%), the maximum pressure on the roof of the tank occurs at the corner.

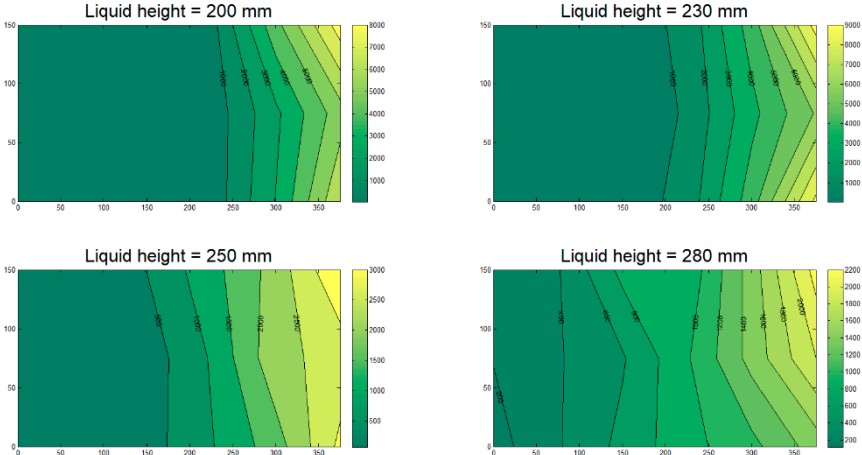

**Figure 7.** Pressure distribution at the roof of the 755 mm × 300 mm tank, 0° orientation.

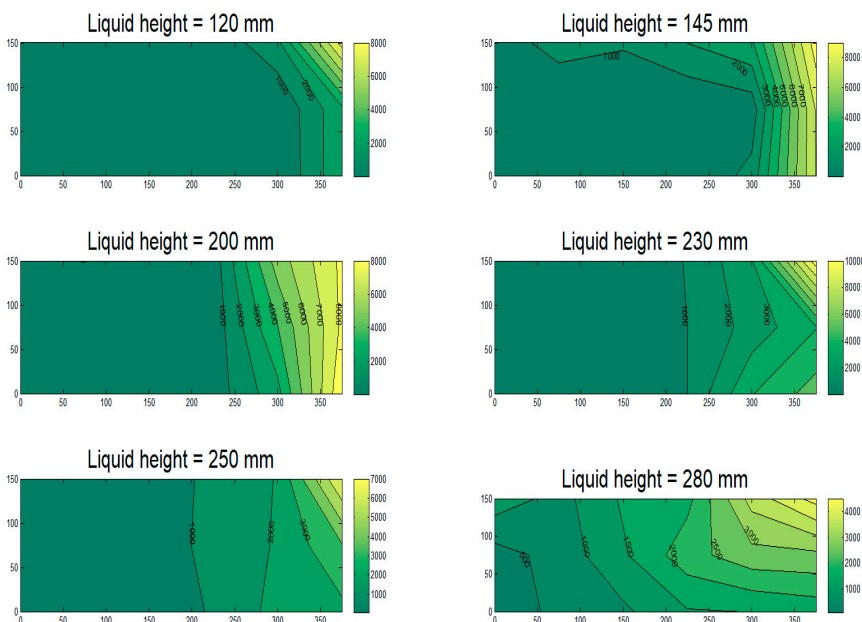

**Figure 8.** Pressure distribution at the roof of the 755 mm × 300 mm tank, 30° orientation.

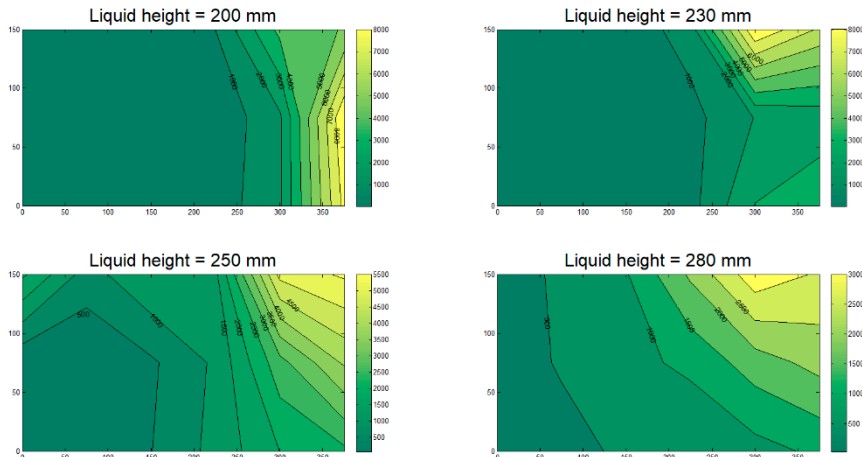

**Figure 9.** Pressure distribution at the roof of the 755 mm × 300 mm tank, 60° orientation.

To find a relationship to predict the maximum pressure for any tank size with any liquid height, the pressure and liquid height need to be dimensionless. It should be noted that the dimensionless maximum pressure needs to consider all factors that might affect the value of the pressure, and hence, the dimensionless pressure and dimensionless liquid height can be calculated by Equations (24) and (25):

$$P_d = \frac{P_{max}}{\frac{(a.\rho.h.L.H)}{(Fb)^2}} \tag{24}$$

$$h_d = \frac{h}{L} \tag{25}$$

where $P_d$ is the dimensionless pressure, $P_{max}$ is the maximum pressure on the roof, $a$ is the maximum acceleration of the harmonic excitation, $\rho$ is the density of water, $h$ is the liquid height in the tank, $H$ is the height of the tank, $L$ is half of the length of the tank (i.e., the

tank's length is 2*L*), and *Fb* is the available freeboard. The parameters *a* and *Fb* can be calculated by Equations (26) and (27):

$$a = A.\omega_i^2 \tag{26}$$

$$Fb = H - h \tag{27}$$

In Equation (26), *A* is the displacement amplitude of the harmonic motion. In Figure 10, the dimensionless maximum pressure plotted against the dimensionless liquid height are presented in a scatter graph. It should be noted that the results presented in this study are valid for cases when the sloshing height exceeds the wall height, which then generates pressure on the roof of a tank.

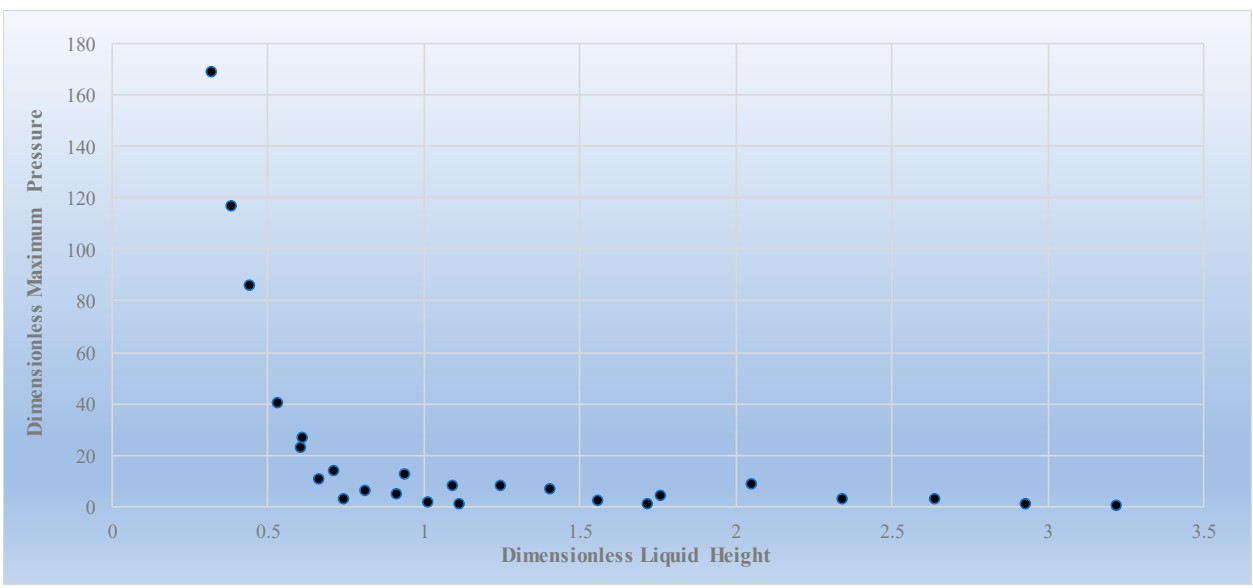

**Figure 10.** Dimensionless maximum pressure versus dimensionless liquid height for the observed (CFD) data.

### 3.2. Genetic Programming

The GPTIPS algorithm allows the user to choose between Single-Gene and Multi-Gene solutions. Single-Gene is the more traditional way of GP and results in simpler equations. Although the Multi-Gene process is more complex, it may lead to solutions with higher accuracy. In this study, the default crossover and mutation coefficients were used as follows: probability of Multi-Gene GP tree cross over = 0.85, probability of Multi-Gene GP tree mutation = 0.1, and probability of Multi-Gene GP tree direct copy = 0.05.

In this section, both Single-Gene and Multi-Gene solutions are examined and explained, and the results are presented.

### 3.2.1. Single-Gene Solution

In the single-Gene solution, the procedure is simple. There is only one gene and a bias term; hence, there is no crossover of sub-trees. Mutations, however, occur in this solution. The equation obtained from the GPTIPS algorithm in the Single-Gene mode is presented in Equation (28):

$$P_{d,\,S} = 4.6489 - \frac{12.498 \times ln(h_d)}{h_d^3 + 0.0534} \tag{28}$$

Here, $P_{d,S}$ is the dimensionless maximum pressure obtained by the Single-Gene solution.

To obtain this equation, the algorithm was set to have 200 generations, with a population size of 300. The maximum tree depth was set to 4, and operators plus, minus, multiply, divide, and log (which in MATLAB means the Napierian logarithm, i.e., ln) were used. This equation is obtained in generation 184. It should be noted that for the simulated tanks,

$h_d$ (i.e., dimensionless liquid height) has a value between 0.3179 and 3.2211, and hence the results are valid for tanks with dimensionless liquid height in that range. Since this relationship is obtained based on the maximum pressure in all tank orientations, it is not affected by the angle of tank orientation. Figure 11 presents the complexity of the model plotted against its accuracy level $(1 - R^2)$ for the population on the training set of data. In this figure, green dots represent Pareto models, and blue dots represent non-Pareto models. The green dot with a red circle shows the best model in terms of $R^2$ on the training data.

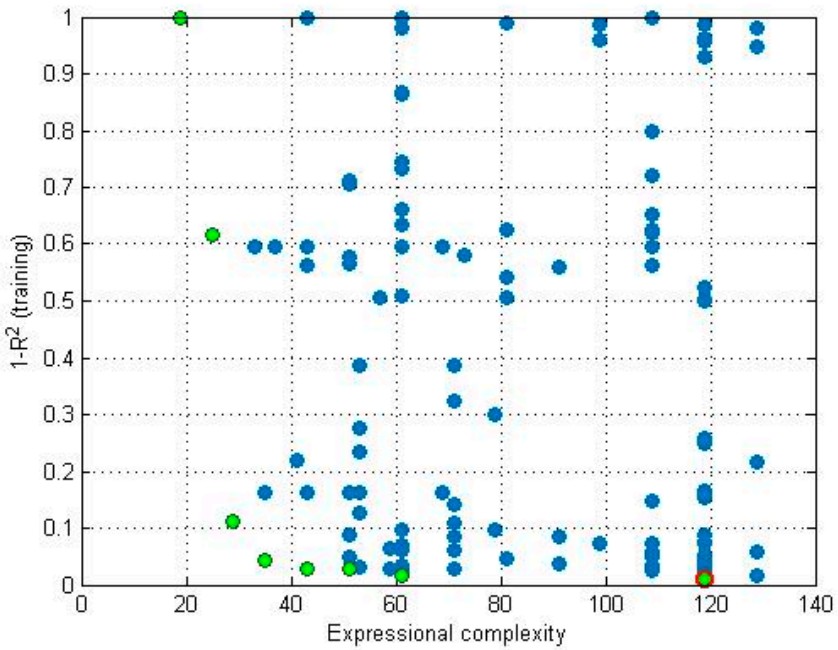

**Figure 11.** Expressional complexity of the proposed Single-Gene model.

3.2.2. Multi-Gene Solution (MGGP)

In this step, the algorithm is modified to use multiple genes. This mode has both crossover and mutation processes. The following equation (Equation (29)) is obtained from the Multi-Gene procedure.

$$P_{d,M} = 5.1961 + \frac{(2.383h_d^3 - 16.846h_d^2 + 17.484h_d - 3.402)}{h_d^5} \tag{29}$$

In this equation, $P_{d,M}$ is the dimensionless maximum roof pressure obtained by the Multi-Gene program. The number of generations was set to 500 with a population of 300. Equation (29) was obtained in generation 473. This equation is composed of the following genes:

$$\text{Gene 1}: \ -\frac{0.920h_d + 3.402}{h_d^5} \tag{30}$$

$$\text{Gene 2}: \ \frac{2.383h_d^2 - 16.85h_d + 18.4}{h_d^4} \tag{31}$$

A bias term equal to 5.196 was obtained. Figure 12 presents the complexity of the model plotted against its accuracy level $(1 - R^2)$ for the population on the training set of data.

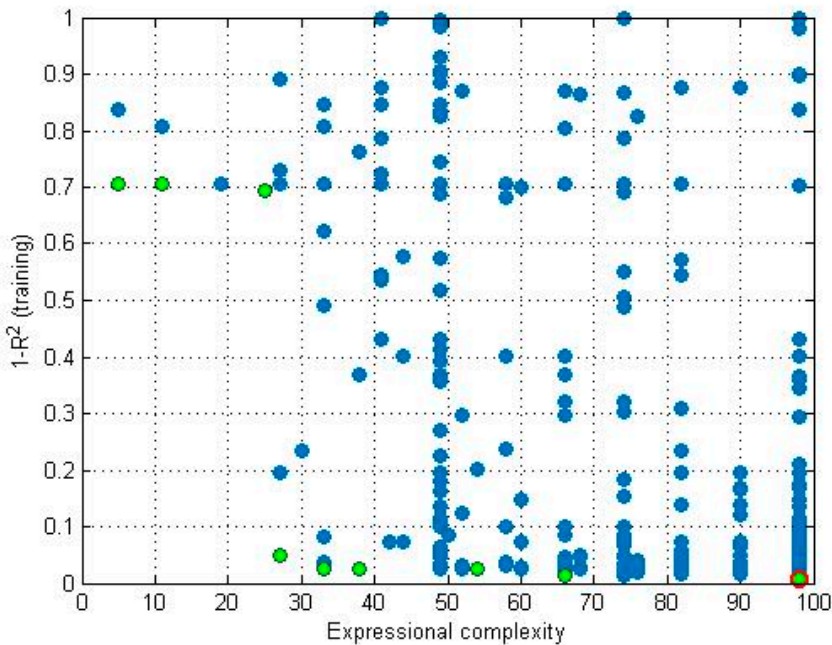

**Figure 12.** Expressional complexity of the proposed Multi-Gene model.

The reason for having a different number of maximum generations for the GP and MGGP models is that for the GP model, the optimum equation was found in the 184th generation, and for the MGGP model it was in the 473rd generation. Therefore, while the 200 maximum generations sufficed for the GP model, the MGGP model required a higher number of maximum generations. These numbers were chosen on a trial and error basis, starting from 100 generations until the optimum equation was obtained at a generation smaller than the maximum number of generations. This could ensure that the obtained equation was the optimal one.

### 3.2.3. Error Estimations

In this section, some error measures of the Single-Gene and Multi-Gene models are presented and compared. These measurements can help determine the accuracy of the presented models and the choice of each option. Errors were measured for both Single-Gene and Multi-Gene programs on the trained and tested data and were finally compared against each other.

a. R-Squared ($R^2$)

In this section, the calculated dimensionless maximum pressure (based on Equations (28) and (29) for Single-Gene and Multi-Gene solutions, respectively) are plotted against the observed dimensionless maximum pressure in Figure 13a,b. The $R^2$, is calculated as

$$R^2 = 1 - \frac{\sum(P_d - P_{d,GP})^2}{\sum(P_d - \overline{P_d})^2} \tag{32}$$

where $P_{d,GP}$ is the dimensionless maximum pressure obtained by the MGGP, and $\overline{P_d}$ is the average of the observed dimensionless maximum pressures. Table 3 presents the $R^2$ values for the Single-Gene and Multi-Gene solutions.

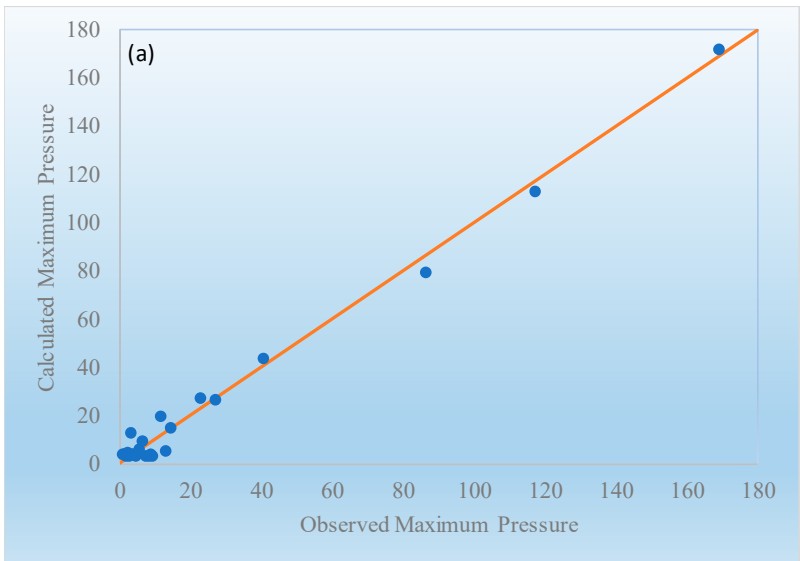

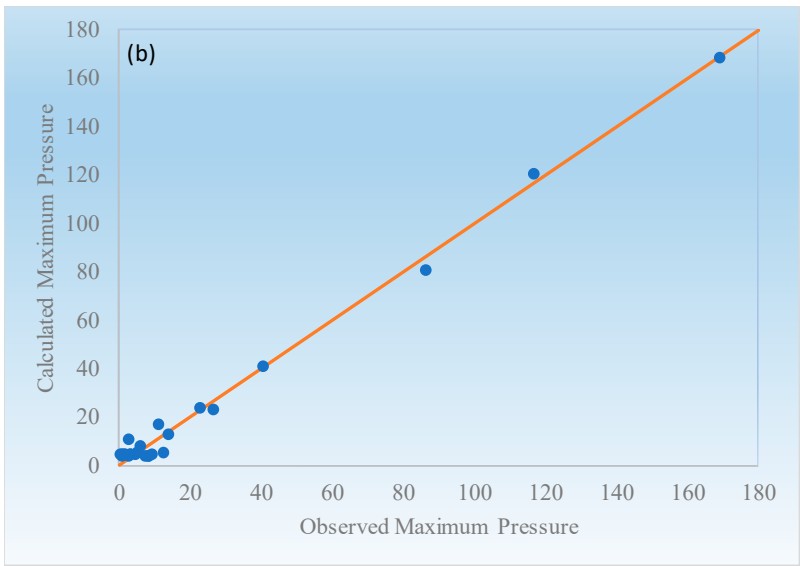

**Figure 13.** Observed dimensionless maximum pressure plotted against the dimensionless maximum pressure obtained by (**a**) Single-Gene procedure and (**b**) Multi-Gene procedure, for the overall data sets.

**Table 3.** MAD measurements.

| Data Set | MAD | | |
|:---:|:---:|:---:|:---:|
| | **Observed Data** | **Single-Gene Results** | **Multi-Gene Results** |
| Trained | 30.63 | 30.27 | 30.44 |
| Test | 6.32 | 17.30 | 5.90 |
| Overall | 26.08 | 25.90 | 25.56 |

b. Root Mean Squared Error (RMSE)

The standard deviation of the residuals, known as root mean squared error (RMSE) is another way of error reporting. It shows the concentration of data near the regression graph. RMSE is calculated based on the following equation:

$$RMSE = \sqrt{\frac{\sum (P_d - P_{d,GP})^2}{N}} \tag{33}$$

where $N$ is the number of observed data, which in this study is 20 for the trained data set, 5 for the test data set, and 25 for the overall data. RMSE has the same dimensions as the original data. In this case, since the input data set is dimensionless, the RMSE is also dimensionless. RMSE values for each of the data sets are presented in Table 4.

**Table 4.** Error estimates.

| Data Set | | R-Squared | RMSE | | | MAE | MAPE (%) | AIC | PI |
| | | | Value | % of Maximum Dimensionless Pressure | % of Mean Dimensionless Pressure | | | | |
|---|---|---|---|---|---|---|---|---|---|
| Single-Gene | Trained | 0.989 | 4.54 | 2.69 | 17.09 | 3.64 | 68% | 21.15 | 0.086 |
| | Test | 0.844 | 3.23 | 14.17 | 46.30 | 3.03 | 260% | 10.55 | 0.241 |
| | Overall | 0.989 | 4.31 | 2.55 | 19.03 | 3.52 | 107% | 23.87 | 0.114 |
| Multi-Gene | Trained | 0.992 | 3.89 | 2.30 | 14.63 | 3.28 | 76% | 21.8 | 0.073 |
| | Test | 0.889 | 2.73 | 11.99 | 39.18 | 2.19 | 302% | 12.18 | 0.202 |
| | Overall | 0.992 | 3.69 | 2.18 | 16.26 | 3.06 | 121% | 24.17 | 0.082 |

c. Mean Absolute Deviation (MAD)

Mean absolute deviation or MAD is a tool for showing the scatteredness of data around the mean. It can be measured by the following equation:

$$MAD = \frac{\sum |P_d - \overline{P_d}|}{N} \text{ or } MAD = \overline{\sum |P_d - \overline{P_d}|} \tag{34}$$

The MAD measurements for each data set are presented in Table 3.

d. Mean Absolute Error (MAE)

Mean absolute error (MAE) is the average of the absolute values of the difference between the observed and measured data. In other words,

$$MAE = \frac{\sum |P_d - P_{d,GP}|}{N} = \overline{\sum |P_d - P_{d,GP}|} \tag{35}$$

The MAE values are presented in Table 4.

e. Mean Absolute Percentage Error (MAPE)

This error measures the accuracy of the model as a percentage and is calculated as follows:

$$MAPE = \frac{1}{N} \sum \frac{P_d - P_{d,\,GP}}{P_d} \times 100\% \tag{36}$$

The MAPE values found in this study for different data sets of Single-Gene and Multi-Gene modes are presented in Table 4.

f.  Akaike Information Criterion (AIC):

The results were also compared using the Akaike information criterion (AIC) using the following equation [43]:

$$\text{AIC} = \text{N} \times \log(\sqrt{\text{RMSE}}) + 2k \tag{37}$$

where k is the number of optimized coefficients. The results are presented in Table 4. The value of the AIC can help compare the complexity and the accuracy of the models at the same time [44]. The results show that when combined, the simplicity and accuracy of the two models (i.e., Single-Gene and Multi-Gene methods) are very close, and there is a difference of 3.1%, 13%, and 1.2% between the Single-Gene and Multi-Gene models for the trained, test, and overall data sets.

g. Performance Index (PI):

In addition to error estimates, evaluating the model performance is helpful in the comparison of different models. The performance index (PI) can be used for this purpose as follows [45]:

$$\text{PI} = \frac{\text{RRMSE}}{\text{R} + 1} \tag{38}$$

$$\text{RRMSE} = \frac{\text{RMSE}}{\left|\overline{\text{P}_d}\right|} \tag{39}$$

$$\text{R} = \frac{\sum(\text{P}_d - \overline{\text{P}_d})(\text{P}_{d,\text{GP}} - \overline{\text{P}_{d,\text{GP}}})}{\sqrt{\sum(\text{P}_d - \overline{\text{P}_d})^2 \sum(\text{P}_{d,\text{GP}} - \overline{\text{P}_{d,\text{GP}}})^2}} \tag{40}$$

where RRMSE is relative root mean square error and R is the correlation coefficient. The lower the PI, the more precise the model. The results of the PI are presented in Table 4. The results show that in all data sets—i.e., test, trained, and overall—the Multi-Gene model has a lower PI, and therefore it is a more precise model than the Single-Gene model.

The error measurements demonstrate that the Multi-Gene method provides a relatively more accurate results compared to the Single-Gene method; however, a rather more complicated formula is required. It is suggested that in the situations where a rough estimate is needed, the Single-Gene method can lead to a reasonable answer in a relatively shorter time with less computational cost, but when a more accurate answer is required, the Multi-Gene formula is recommended.

The error estimates show that the test data sets in both Single-Gene and Multi-Gene models have a lower $R^2$ and higher MAPE, which can be indicators of higher errors and overfitting of the model. However, the RMSE and MAE values provide comparable results for the test and trained data sets with fewer errors. In other words, two of the four error indicators show better results in test data sets, while the other two may indicate overfitting. Given the circumstances, the results for both Single-Gene and Multi-Gene models are reasonably acceptable.

### 3.3. Uncertainty Analysis and Confidence Bands

After finding the equation, its credibility needs to be investigated and verified by uncertainty and sensitivity analyses.

A Monte Carlo analysis was also performed for the uncertainty analysis of the resulting equation. The objective of this analysis is to calculate the uncertainty of the final function. To do so, 1,000,000 random inputs of $h_d$ were generated in the range of 0.3179 to 3.2211. Then, the equation was run for each random number. To generate random data with normal-shaped distribution in a specific range, a truncated Gaussian function was used. The histogram of the generated data using the truncated Gaussian function is shown in Figure 14.

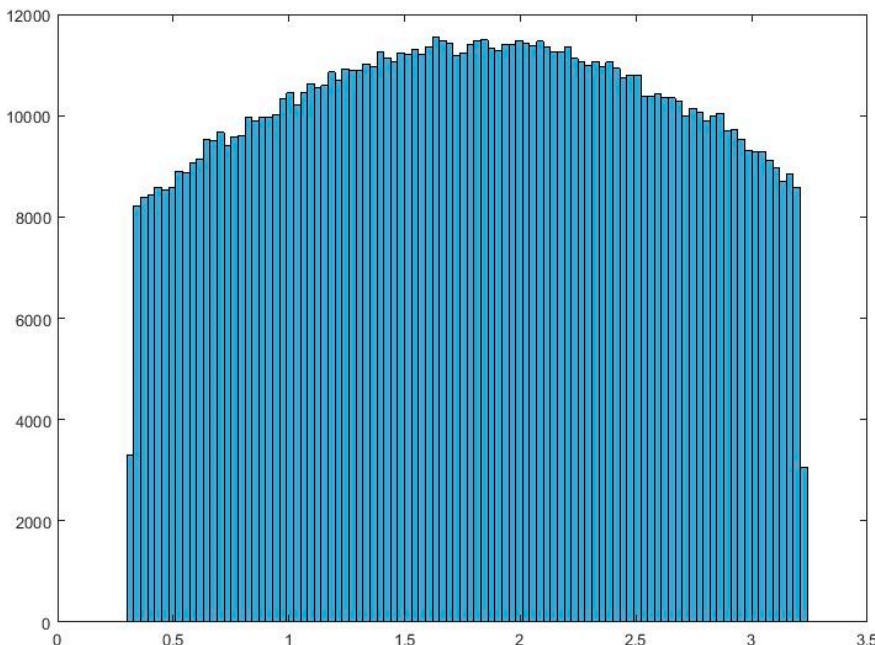

**Figure 14.** Histogram of generated random inputs created with truncated Gaussian function.

These random numbers were then put into the GP model, and 1,000,000 values for $P_d$, namely $P_{mc}$, were calculated. The mean absolute deviation (MAD) was calculated around the average using Equation (27)

$$MAD = \frac{1}{n} \sum_{i=1}^{n} \left| P_{mc_i} - P_{avg} \right| \qquad (41)$$

where $n$ is the number of samples (i.e., $n$ = 1,000,000 in this case) and $P_{avg}$ is the average of the pressures calculated by the Monte Carlo simulation [20], thus leading to

$$MAD_{SG} = 11.718 \text{ and } MAD_{MG} = 11.4728$$

This can be used to calculate the uncertainty percentage of the function by using the following equation [20]:

$$U = 100 \times \frac{MAD}{P_{avg}} \qquad (42)$$

The above leads to

$$U_{SG} = 100 \times \frac{11.718}{10.7485} = 109.02 \text{ and } U_{MG} = 100 \times \frac{11.4728}{11.430} = 100.3738$$

where $U_{SG}$ and $U_{MG}$ are the uncertainty percentages for the Single-Gene and Multi-Gene equations, respectively. Due to the high slope of the graph of the equation in the beginning, these amounts of uncertainty are reasonable.

Confidence bands of the graph are then obtained using a 2nd-order approach in the calculation of the Jacobian Matrix with the central difference scheme. The MATLAB internal function "nlpredci" (non-linear regression prediction confidence intervals) is used. This function can provide the user with 95% confidence band widths of the given equation. According to Dolan et al. [46], this function gives a symmetric confidence interval at each point; hence, the two confidence bands have the same distance from the main equation. The 95% confidence bands for Equations (28) and (29) are plotted in Figure 15a,b, respectively. The average confidence band width for Equation (28) (i.e., Single-Gene mode) is 20.54, and for Equation (29) (i.e., Multi-Gene mode) is 15.27.

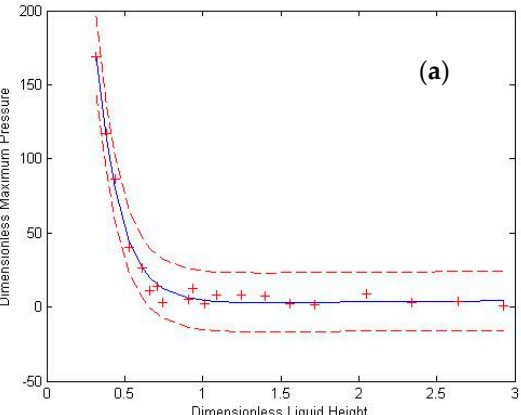
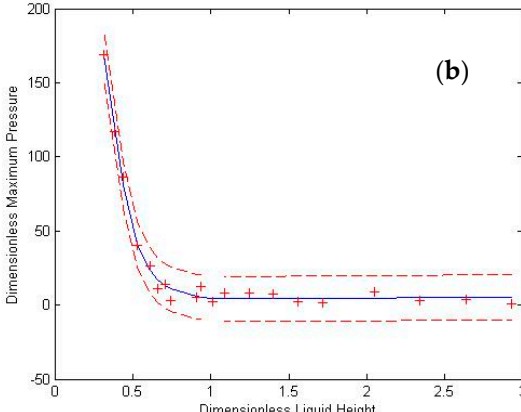

**Figure 15.** Graph of the proposed equation for dimensionless pressure plotted against dimensionless liquid height with 95% confidence bounds for (**a**) Single-Gene and (**b**) Multi-Gene modes.

### 3.4. Sensitivity Analysis

For the sensitivity analysis, a 10% perturbation is applied to an input value of the equation (here, the mean), and the perturbation in the outcome is calculated. The calculations are presented in Equations (43)–(45):

$$h_{dp} = 1.1 \times h_{dm} \tag{43}$$

$$\Delta P_d = \frac{\left| P_{dp} - P_{dm} \right|}{P_{dm}} \tag{44}$$

$$S_n = \frac{\Delta P_d}{0.1} \tag{45}$$

where $h_{dp}$ is the 10% perturbed mean dimensionless liquid height, $h_{dm}$ is the actual mean dimensionless liquid height, $\Delta P_d$ is the perturbation that appears in the dimensionless pressure due to the 10% perturbation in the dimensionless liquid height, $P_{dp}$ is the change in the value of the dimensionless pressure when the dimensionless liquid height changes, $P_{dm}$ is the value of the dimensionless pressure at mean dimensionless liquid height ($h_{dm}$) calculated based on Equations (28) and (29) for Single-Gene and Multi-Gene modes, and $S_n$ is the normal sensitivity of those equations.

This leads to a sensitivity of $S_{n,SG} = 0.258$, or a 25.8% sensitivity for the Single-Gene solution and $S_{n,MG} = 0.116$ or a 11.6% sensitivity for the Multi-Gene solution.

## 4. Conclusions

The purpose of this study was to develop an empirical equation for the maximum pressure at the roofs of liquid storage tanks. To do so, a previously validated OpenFOAM model was used to generate the data. The data included the maximum pressure at the roof. Various tank sizes with different liquid heights were modeled, and harmonic sinusoidal base excitations with resonance frequencies were applied to the tanks. To consider the effect of bi-directional excitation, the tanks were shaken in three different orientations. Pressure sensors were distributed on one quarter of the roof, and the maximum pressure at each sensor was recorded.

Using the GP method, a relationship between the dimensionless liquid height and the dimensionless maximum pressure was obtained in both Single-Gene and Multi-Gene modes (Equations (28) and (29)). Using multiple error measures, the two equations were tested, and the results were compared. These results show that the outputs of the equations are in good agreement with the ones obtained by CFD modelling. Uncertainty analyses of the equations were conducted using the Monte Carlo method, leading to reasonable values

given that both functions have an ascending shape with a high slope in the beginning of their domains. In addition, the 95% confidence bands for the equation were drawn.

It can be concluded that the use of AI techniques combined with CFD is helpful in predicting the maximum pressure at the roof of a base-excited tank. Further investigation on this aspect is currently in progress by the authors.

**Author Contributions:** Conceptualization, A.M. and R.K.; data curation, I.B.T.; formal analysis, I.B.T.; funding acquisition, A.M. and R.K.; investigation, I.B.T.; methodology, A.M. and R.K.; project administration, A.M. and R.K.; resources, I.B.T.; software, I.B.T. and A.M.; supervision, A.M. and R.K.; validation, I.B.T.; visualization, I.B.T.; writing—original draft, I.B.T.; writing—review and editing, A.M. and R.K. All authors have read and agreed to the published version of the manuscript.

**Funding:** This research was funded by Natural Sciences and Engineering Research Council of Canada (NSERC).

**Conflicts of Interest:** The authors declare no conflict of interest.

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
