# Peer review of "Prediction of Maximum Pressure at the Roofs of Rectangular Water Tanks Subjected to Harmonic Base Excitation Using the Multi-Gene Genetic Programming Method"

_mca, doi:10.3390/mca26010006_

Round 1

Reviewer 1 Report

The authors applied multi-genetic genetic programming to predict maximum pressure at the roofs of rectangular water tanks subjected to harmonic base excitation. I am left to believe that reviews needed are above major reviews and that the article needs considerable reworking to be accepted. I think the paper can be accepted in the current Journal and recommend for publication after major revision. However, some items should be considered before more assessment. I would recommend to the authors:

(1) In the Introduction section, the main contribution of the current study should be provided in detail.

(2) The authors must be provided the main gap of the previous study in the prediction of maximum pressure at the roof of the rectangular water tanks that overcome in the current study.

(3) According to that the GEP can provide the final model with different genes, what is the main difference between the GEP and MGGP?

(4) It is necessary to compare the results of MGGP, GP, and GEP in terms of accuracy and simplicity, simultaneously, using the Akaike Information Criterion provided in the following references:

https://link.springer.com/article/10.1007/s00707-017-2043-9

https://link.springer.com/article/10.1007/s00704-018-2436-2

(5) The authors stated that “For this study, 80% of the data is used to train the model and the rest is used for testing (e.g., model validation)”. I cannot find the number of all samples. 80% is accepted for training when the total amount of data is sufficient. According to Figure 10 and Table 2, I think all samples are 25. Therefore, the authors must be applied to k-fold cross-validation.

(6) The manuscript could be substantially improved by relying and citing more on recent literature about contemporary real-life case studies of genetic-based approaches Following, you will find some new related references which should be added to the literature review.

https://www.sciencedirect.com/science/article/abs/pii/S1001627917301270

https://link.springer.com/article/10.1007/s00521-018-3411-7

(7) The authors stated that “To obtain this equation, the algorithm was set to have 200 generations, with a population size of 300”. Please provide more details about how you choose them as well as maximum tree depth and operators. Arbitrary or through a trial and error process. I cannot find any results for different efforts to find the optimum value for each of them.

(8) Please proved more details about Figures 11 and 12. It is not clear what information does this figure provides? What are the blue and green circles? There is no legend.

(9) What the number of generation in the GP (i.e. 200) and MGGP (i.e. 500) is not equal? A higher number of generations may lead to higher performance. Please clarify it.

(10) What are the mutation and crossover coefficients for MGGP?

(11) Section 3.3: All of the applied indices must be defined in the previous sections. They are not results. Besides, the results of them should be provided in a table. Tables 3-7 should be present in a Table.

(12) There is no information about Figure 13 that provides for training, testing, or overall samples.

(13) I think overfitting occurred. In Table 3, for both single and multi-gene, the R-Squared in the test is 10% lower than the train. In Table 7, the MAPE for the test is more than four times the value of this index at the training stage (for both single and multi-gene).

Reviewer 2 Report

This paper shows us an artificial intelligence method that was used to solve the sloshing problem. We can see that the author has done a lot of work around this topic, such as the empirical expressions of the maximum sloshing pressure on the tank roof, sloshing pressure in different sloshing tanks, the sensitivity and uncertainty of the proposed empirical equation, and so on. This topic should be very useful in the field of engineering, and it is also very attractive to readers.

While, this paper, in the present version, may need major revisions to be improved for publication. There are several aspects as follows:

  1. Introduction
  • the author said that the sloshing phenomenon will cause structural failure and spillage of the liquid into the environment. Actually, most tank damage does not just occur at the roof. We need sufficient proof to support the importance of the author’s work on the pressure analysis of the tank roof.
  • In the introduction, the seismic excitation was mentioned as the external excitation. While a harmonic base excitation was used in the present research. Because seismic excitation and simple harmonic excitation are very different in time and frequency domain. And the work in this article does not seem to be sensitive to excitation. Based on the background of this article and the numerical model used by the author, why not use seismic signals as excitation? I recommend that the author use at least one seismic signal as excitation to test whether the empirical equation given now is still valid.
  • I don't think this paragraph is necessary for this article, Lines 55-58.
  • The introduction about the Genetic programming methods has not been accurately introduced. There are too many basic concepts here, and the relationship between the above methods and engineering issues is not properly discussed.
  • The full name of OpenFOAM should be given for the first time in Line 91.
  1. Materials and Methods
  • You may not be able to use a paper under review as a valid basis for your model in Line 103. One available way is to give a simple verification case in this article, and the other way is to cite related work of other scholars (Sloshing motion in a real-scale water storage tank under nonlinear ground motion. Water, Fluid dynamics analysis of sloshing pressure distribution in storage vessels of different shapes. Ocean Engineering, ……)
  • Regarding the natural frequency of the tank, you may be able to use the formula of Lamb (1932). The expression and use of this formula are simpler. (Lamb, H. (1932). Hydrodynamics. Cambridge university press.)
  • I cannot understand how the variable i in Eq. 1 is reflected on the right side of the equation.
  • As a 3D simulation was conducted in this research and a rectangular tank was used. The tank sizes in orientations of 0° and 90° were different. You may need to add another case or you may need to use a cube tank.
  • The introduction of the turbulence model of the present simulation should be given.
  • Any value between the two shows a partially filled particle, which is an indicator of the free surface.” Actually, the principle that we usually choose the free surface is to choose a value close to 0.5 for the fluid volume fraction. If you use all values between 0 and 1, it will cause the free surface to be rough, and a lot of splashing liquid will also be considered as a part of the free surface.
  • In this article, OpenFOAM is just a numerical model that is used to provide basic data. The core content is artificial intelligence algorithms. As a model, your introduction of OpenFOAM is a bit wordy.
  • The word “tracked” in Line 162 may not correct.
  • Table 2 in Line 163: The height of the liquid may be expressed by a dimensionless number. In this way, it is difficult for readers to understand the meaning of the selected height under different tank sizes.
  • As a 3D tank was used. And take a look at your results in Figure 8, the pressure sensors on the 1/4 roof may not be enough to express the pressure characteristics of the entire tank top.
  • The introduction of the method of the single-gene model was missing in section 2.2.
  • Results and discussion
  • In Fig. 10, the results show that the shallower the water, the greater the pressure. While if the height of the liquid was very shallow. There may be no pressure on the roof, and such a height of the water level may have to be given or discussed.
  • In Tables 4 – 7, necessary discussion about the results may be necessary.
  1. Conclusion
  • The conclusions need to be further summarized and refined.

Reviewer 3 Report

Prediction of Maximum Pressure at the Roofs of Rectangular Water Tanks Subjected to Harmonic Base Excitation Using the Multi-Gene Genetic Programming Method

This is an interesting paper which illustrates the capability of genetic programming to formulate the Prediction of Maximum Pressure at the Roofs . New models are presented in terms of several parameters. The article is well organized and the idea is expressed clearly. The paper falls within the scope of journal. However they should address the following comments to meet the journal standards.

1. The literature review of GEP applications in civil engineering should be expanded and the main references in this field should be cited such as:
* "Handbook of Genetic Programming Applications"
* "Formulation of shear strength of slender RC beams using gene expression programming"
* "Novel Approach to Strength Modeling of Concrete under Triaxial Compression"

2. The English of the paper is good. But some typographical mistakes need to be corrected. Sections need numerical counter

3. The models should be compared using a suitable criterion not only using R or error functions. It is because R will not change significantly by shifting the output values of a model equally, and error functions (e.g. RMSE, MAE and MAD) only shows the error not correlation. Therefore, the criteria should be combination of R, RMSE and/or MAE, MAD. Add other criteria such as performance index (PI) ["Assessment of artificial neural network and genetic programming as predictive tools" Advances in Engineering Software] to compare all models in table 4.

4. Compute the sensitivity values using a general method like the method presented at "An evolutionary approach for modeling of shear strength of RC deep beams. Materials and Structures".

Round 2

Reviewer 2 Report

After revision, I agree to publish the current version of the manuscript.

Reviewer 3 Report

Revised version is acceptable